# Visual SKETCHPAD: Sketching as a Visual Chain of Thought for Multimodal Language Models

**Yushi Hu**[*1,2]   **Weijia Shi**[*1]   **Xingyu Fu**[3]   **Dan Roth**[3]   **Mari Ostendorf**[1]
**Luke Zettlemoyer**[1]   **Noah A. Smith**[1,2]   **Ranjay Krishna**[1,2]

[1]*University of Washington*   [2]*Allen Institute for AI*   [3] *University of Pennsylvania*

## Abstract

Humans draw to facilitate reasoning: we draw auxiliary lines when solving geometry problems; we mark and circle when reasoning on maps; we use sketches to amplify our ideas and relieve our limited-capacity working memory. However, such actions are missing in current multimodal language models (LMs). Current chain-of-thought and tool-use paradigms only use text as intermediate reasoning steps. In this work, we introduce SKETCHPAD, a framework that gives multimodal LMs a visual sketchpad and tools to draw on the sketchpad. The LM conducts planning and reasoning according to the visual artifacts it has drawn. Different from prior work, which uses text-to-image models to enable LMs to draw, SKETCHPAD enables LMs to draw with lines, boxes, marks, etc., which is closer to human sketching and better facilitates reasoning. SKETCHPAD can also use specialist vision models during the sketching process (e.g., draw bounding boxes with object detection models, draw masks with segmentation models), to further enhance visual perception and reasoning. We experiment on a wide range of math tasks (including geometry, functions, graph, chess) and complex visual reasoning tasks. SKETCH-PAD substantially improves performance on all tasks over strong base models with no sketching, yielding an average gain of 12.7% on math tasks, and 8.6% on vision tasks. GPT-4o with SKETCHPAD sets a new state of the art on all tasks, including $V^*$Bench (80.3%), BLINK spatial reasoning (83.9%), and visual correspondence (80.8%).All codes and data are in https://visualsketchpad.github.io/.

## 1   Introduction

Sketching is a fundamental human activity, serving as a versatile tool for communication [11], ideation [48], and problem-solving [47]. Unlike written language, sketches have the advantage of conveying visuo-spatial ideas directly, for example by using spatial relations on paper to convey spatial relations or other more abstract relationships in the world. This may explain their ubiquity; maps [40] and architectural plans [12] have been found incised in stone, etched on leather, impressed in clay, and drawn on paper in diverse cultures scattered across the world [39]. Sketches are so fundamental that we use them to teach school children how to solve geometry problems by drawing support lines, to aid engineers conveying prototypes, to support architects creating blueprints, and to allow scientists like us to convey scientific contributions (see Figure 1).

As multimodal language models (LMs) [35, 42, 28, 27, 1, 3, 49, 7, 31, 41, 6, 5] have begun to mature, we now expect them to solve tasks like the ones mentioned above, i.e., ones where people draw intermediate sketches to simplify reasoning. Popular benchmarks now include questions

---

*Equal Contribution. Correspond to <Yushi Hu: yushihu@uw.edu>, <Weijia Shi: swj0419@uw.edu>

38th Conference on Neural Information Processing Systems (NeurIPS 2024).

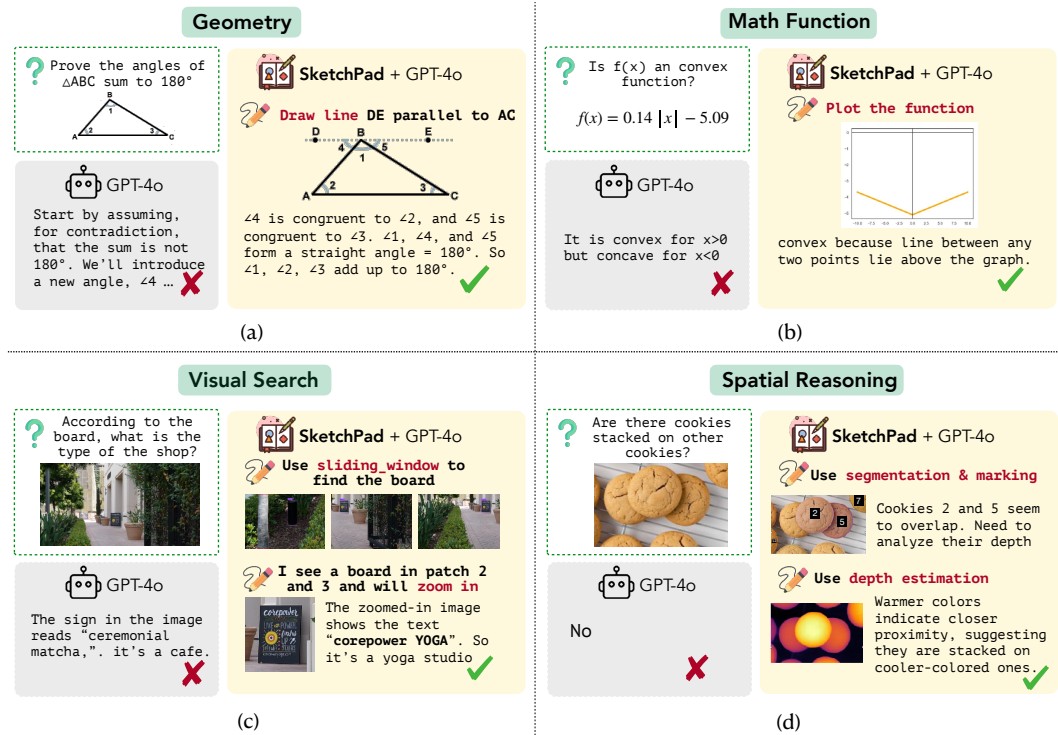

Figure 1: **SKETCHPAD equips GPT-4 with the ability to generate intermediate sketches to reason over tasks.** Given a visual input and query, such as proving the angles of a triangle equal 180°, SKETCHPAD enables the model to draw auxiliary lines which help solve the geometry problem. The examples are from [8, 51, 44]. For all these examples, without SKETCHPAD, GPT-4o fails to get the correct answer, while SKETCHPAD + GPT-4o achieves the correct solution.

about geometry (e.g., Geometry3K [32]) and complex math problems (e.g., IsoBench [8]). In these benchmarks, models are given images of diagrams and asked questions requiring symbolic grounding and spatial understanding, where intermediate sketches like auxiliary lines can enhance reasoning. Even benchmarks in computer vision now have a similar flavor. Specialist vision models can be viewed as sketching on natural images. For example, object detection is plotting bounding boxes around objects; depth estimation is drawing colormaps according to depth. The recently proposed BLINK benchmark [9] would benefit significantly from such intermediate visual sketches. Similarly, the *V*Bench benchmark [51] could focus reasoning on image crops to find answers. Unfortunately, current LMs lack a scaffold for using sketch-based reasoning when solving tasks.

In this paper, **we introduce Visual SKETCHPAD: a framework that provides multimodal LMs with the tools necessary to generate intermediate sketches to reason over tasks.** Inspired by textual chain-of-thought reasoning in LMs [50, 62], SKETCHPAD prompts the underlying visual LM to produce visual artifacts as part of a chain of mixed textual, programmatic, and visual reasoning. For example, to prove that the angles of triangles sum up to 180 degrees in Figure 1 (a), SKETCHPAD enables agents to modify the diagram by introducing a new auxiliary line. This new line, along with new annotated angles, provides the critical information to solve the geometry task. Similarly, SKETCHPAD improves models' spatial reasoning for computer vision. To determine if there are cookies stacked on top of other cookies in the image (Figure 1b), the model first produces an intermediate depth estimate. By analyzing the depth estimate, which reveals cookies overlapping at different depths, the model is able to correctly answer that the cookies are indeed stacked.

We demonstrate the effectiveness of visual SKETCHPAD across a wide range of mathematics and computer vision tasks. For math, we tackle problems including (1) geometry [32], (2) mathematical functions, (3) graph algorithms, and (4) strategy games [8]. For geometry questions, SKETCHPAD enables models to generate Matplotlib code with auxiliary lines and variables, given the diagram input and questions (Figure 1a). Notably, even when the input is solely language-based, such as mathematical functions, SKETCHPAD enables models to plot the functions and reason about their properties, using only the mathematical function expression as input (Figure 1b). These results

highlight the ability of SKETCHPAD to aid reasoning, even in tasks with purely language-based inputs. **Across all four categories of mathematical tasks, SKETCHPAD consistently improves the baseline GPT-4o performance, yielding an average gain of 11.2%.** For computer vision, we tackle diverse tasks including (1) depth, (2) spatial reasoning, (3) jigsaw, (4) visual correspondence, (5) semantic correspondence, as well as questions from (6) the MMVP and (7) the *V*\*Bench benchmarks [9, 44, 51]. For this domain, SKETCHPAD enables models to generate segmentation masks, crop images, draw bounding boxes, zoom into image regions, overlay images, etc. Similar to math, **SKETCHPAD shows consistent improvements across all seven types of computer vision tasks**. For example, GPT-4o, augmented with SKETCHPAD, sees 14.3% improvement on *V*\*Bench, 12.1%, and 9.7% improvements on BLINK's depth and semantic correspondence tasks, setting a new state of the arts across all tasks. Finally, we analyze the effectiveness of SKETCHPAD by comparing the plans generated by our model with human-created plans, showing that they are well-aligned and exhibit similar reasoning patterns. We hope SKETCHPAD opens up new research opportunities toward more capable and interpretable multimodal intelligence.

## 2   Related Work

SKETCHPAD generalizes recent work on multimodal tool-use and visual prompting. We also place our work within the larger sphere exploring LMs as agents.

**Visual programming and tool-use.** With the advancement of LMs [4, 35, 42, 45, 13], researchers have demonstrated the possiblity of decomposing complex vision tasks into simpler substeps that can each be solved using vision tools [57, 60, 18, 17]. Among them, the most relevant to us are Visprog [14] and ViperGPT [38]. They use LMs to generate Python code, which sequentially invokes specialized vision tools. These methods share a common problem that the multimodal modules follow a pre-defined plan outlined by the LM. By contrast, SKETCHPAD allows LMs to change their plan according to the intermediate visual artifacts, yielding better performance and robustness when solving complex multimodal tasks.

**Visual prompting.** Recent work shows that multimodal models can be augmented by visual prompts added to natural images [37]. For example, SoM [55] shows that adding labeled segmentation masks on images unleashes GPT-4V's visual grounding ability. Prior work also reports similar findings in 3D [25] and Robotics [34]. SKETCHPAD is a generalized framework for all these methods, allowing LMs to decide what visual prompting to use as part of the multimodal reasoning process.

**LMs as agents.** Recent work has started to treat LMs as agents that can both reason and act [58, 33, 52, 36, 59]. Researchers have applied this idea to software engineering [20, 61, 15], robotics [34], vision [29, 57], and GUI navigation [54, 23, 53]. SKETCHPAD can also be viewed as an agent that accepts multimodal inputs and outputs. One big difference is that SKETCHPAD can create visual artifacts to facilitate reasoning, while prior LM agents only generate text during reasoning.

## 3   Visual SKETCHPAD

We introduce visual SKETCHPAD, a general framework that enables multimodal LMs to draw sketches as intermediate reasoning steps and to use these sketches to facilitate further reasoning. Figure 2 shows examples of how SKETCHPAD works. Given a multimodal query, SKETCHPAD agent generates a sketching plan to address the query (*Thought*), and then synthesizes a program to create visual sketches (*Action*). By analyzing the resulting sketches (*Observation*), which serve as a visual representation of the reasoning process, the model generates a final response to the query.

Our framework requires no finetuning or training. Multimodal LMs, out of the box, can be prompted to sketch using our framework. Our implementation is based on the AutoGen [52] framework. We give the overview of our SKETCHPAD framework in §3.1, and delve deep into how it integrates sketching into the reasoning process in §3.2.

### 3.1   Overview of SKETCHPAD

The SKETCHPAD agent solves tasks by engaging in an iterative interaction process with an environment. Given a multimodal query $q$ that includes both visual and textual components, the model generates a series of thoughts, actions, and observations to gather the information needed to answer the query. At each time step $t$, the model performs three key steps:

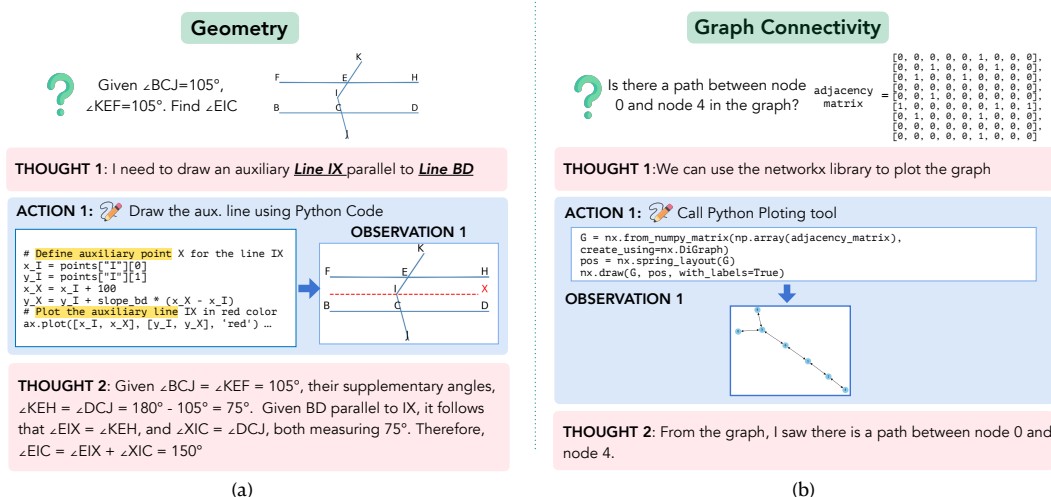

Figure 2: **Overview of SKETCHPAD**. Given a multimodal query, the SKETCHPAD agent generates a sketching plan to address the query (*Thought*), and then synthesizes a program to create visual sketches (*Action*). By analyzing the resulting sketches (*Observation*), which serve as a visual representation of the reasoning process, the model generates a final response to the query.

**Thought:** The model analyzes the current context $c_t$, which includes the query, previous thoughts, actions, and observations, to generate a thought plan $p_t$ for the next action. For example, given the query $q$ – "*find the $\angle EIC$*" in Figure 2a, the model's thought plan $p_1$ is to draw an auxiliary line $IX$ parallel to $BD$ serving as a ***visual sketch*** to help solve the problem.

**Action:** Based on the thought plan, the model executes an action $a_t$, which can manipulate both visual and textual content. In the geometry example, to realize the proposed thought of drawing the auxiliary line, the model generates Python code to modify the original geometry diagram. The generated code is then compiled and executed.

**Observation:** Based on the action $a_t$, SKETCHPAD's environment returns a new observation $o_{t+1}$, such as a new diagram with the auxiliary line drawn in the geometry example. The multimodal context is then updated to $c_{t+1} = (c_t, p_t, a_t, o_{t+1})$.

The multi-turn interaction process continues until time step $T$, when the model determines that it has gathered enough information from the context $c_T$ to answer the query. At this point, it generates a special **Terminate** action and provides the answer.

Different from prior work [58], where LMs primarily generate and manipulate text-based observations and actions, SKETCHPAD enables the model to work with **multimodal observations $o_t$ and actions $a_t$, manipulating both visual and textual content.** This allows the model to plan and reason with the visual sketches it has drawn, enhancing its problem-solving capabilities.

## 3.2 Sketching via Code Generation

The core component of SKETCHPAD is sketching, which enables the LM to generate visual sketches by synthesizing programs that call different specialist vision models or Python plotting packages.

**Program Generation.** Similar to recent works like ViperGPT and VPD [14, 38, 18], SKETCHPAD enables LMs to sketch through code generation. The LM is provided, through a prompt, with a detailed description of the available tools that can generate multimodal content (an example prompt and description can be found in §C). The prompt includes Python function signatures and docstrings [16] for these modules, but does not contain their full implementation. The LM generates Python code in a code block, using the provided tools, which, when executed, generates new image and text outputs. A special *display* function allows the LM to **visualize** the sketch image in the next observation $o_{t+1}$.

**Modules for sketching.** SKETCHPAD uses various tools to facilitate the sketching process, depending on the task at hand. For mathematical tasks, SKETCHPAD uses common Python packages like

`matplotlib` and `networkx` for plotting (see §4). For vision tasks, the LM leverages **specialist vision models** during the sketching process. These models include detection tools that draw bounding boxes on the image, as well as segmentation and marking tools (inspired by SoM [55]) that draw colorful masks on the image and use numbers to label segments. We find these specialists possess useful perception skills for visual reasoning tasks, and SKETCHPAD is an effective way to combine them into a multimodal LM (see §5.1).

## 4 Sketching to Solve Math Problems

In this section, we experiment with SKETCHPAD on four complex mathematical tasks : (1) geometry, (2) mathematical functions, (3) graph algorithms, and (4) game strategies. We demonstrate that incorporating sketching capabilities into LMs significantly improves their performance on these mathematical problems, setting new state-of-the-art results (§4.1).

Details of our evaluation tasks and the tools employed for visual reasoning are as follows:

**Geometry Problems.** Drawing auxiliary lines in geometry diagrams is often helpful for problem-solving. For example, in Figure 2 (a), when asked to find $\angle EIC$, the LM plans to draw an auxiliary line $IX$ parallel to $BD$, allowing it to use the properties of parallel lines to determine $\angle EIC$. To evaluate the effectiveness of SKETCHPAD, we use the problems from the Geometry3K dataset [32].

To realize the line drawing process, SKETCHPAD takes a geometry diagram and its corresponding `matplotlib` code as input. The model then proposes and modifies the code to generate auxiliary lines, and executes the modified code to visualize the updated diagram with the added lines.

**Mathematical functions.** Understanding mathematical functions is crucial for various applications in science, engineering, and economics. We focus on two tasks related to mathematical functions from the IsoBench datasets [8]:

- ***Classifying parity*** aims to determine whether a function is even, odd, or neither. Even functions satisfy $f(-x) = f(x)$ for all $x$, while odd functions satisfy $f(-x) = -f(x)$.
- ***Identifying convexity/concavity*** aims to determine whether a function is convex or concave.

Existing LMs can only analyze functions and attempt to prove their properties analytically. [1] However, SKETCHPAD enables them to visually sketch functions to solve problems more efficiently. For instance, to determine the convexity of the function in Figure 1b, SKETCHPAD allows the model to plot the function using `matplotlib`, and visually inspect its overall shape.

**Graph algorithms.** Many real-world problems, such as those related to computer networks and transportation systems, can be formulated as graph problems. We evaluate SKETCHPAD on three graph problems from IsoBench [8]:

- ***Graph connectivity*** determines whether there exists a path between two vertices in a graph.
- ***Maximum flow*** aims to find the maximum amount of flow that can be sent through a network from a source vertex to a sink vertex, subject to capacity constraints on the edges.
- ***Graph isomorphism*** tests whether two graphs are structurally equivalent.

Given an adjacency matrix of a graph like in Figure 2(b), SKETCHPAD can draw the actual graph structure, using using Python's `networkx` library, enabling direct visual reasoning about graph properties and relationships.

**Game strategies.** Chess games can be represented in various formats, including visual board states and textual move notations. Given only the textual move notations, SKETCHPAD can draw the visual representations of the chess board to analyze positions and formulate strategies. We evaluate the performance of SKETCHPAD on the winnder identification task from the IsoBench datasets [8] that aims to find the outcome of a chess game (win for White, win for Black, or draw) based on the final board state. To create the graphical board, SKETCHPAD uses Python's `chess` library to draw the board from the Forsyth-Edwards Notation (FEN) of chess.

---

[1]For humans, the analytical approach is the correct way to tackle these math tasks. However, we observe that LMs are not good at analytical reasoning in math. They make errors when deducing $f(-x)$ and derivatives.

| | Geometry | Graph | | | Math | | Game |
|---|---|---|---|---|---|---|---|
| Model | Geometry | Maxflow | Isomorphism | Connectivity | Convexity | Parity | Winner ID |
| *Prior LLMs without visual inputs* | | | | | | | |
| Gemini-Pro | \ | 15.6 | 47.7 | 50.0 | 87.9 | 48.2 | 8.1 |
| Claude 3 OPUS | \ | 56.3 | 50.0 | 82.0 | 93.0 | 77.6 | 74.4 |
| Mixtral 8x7B [19] | \ | 8.6 | 50.0 | 62.5 | 69.1 | 41.7 | 7.4 |
| LLaMA-2-70B [46] | \ | 18.0 | 50.0 | 50.0 | 74.2 | 33.3 | 12.4 |
| *Latest multimodal LLMs + Visual Sketchpad* | | | | | | | |
| GPT-4 Turbo | 37.5 | 32.8 | 62.5 | 66.0 | 57.0 | 80.5 | 50.4 |
| + Sketchpad | 45.8 | 96.8 | 97.6 | 97.6 | 77.3 | 71.5 | 64.2 |
| | +8.3 | +64.0 | +35.1 | +31.6 | +20.3 | -9.0 | +13.8 |
| GPT-4o | 62.5 | 25.0 | 50.8 | 96.1 | 87.2 | 84.4 | 61.1 |
| + Sketchpad | **66.7** | **66.3** | **65.3** | **98.4** | **94.9** | **94.7** | **64.6** |
| | +4.2 | +41.3 | +14.5 | +2.3 | +7.7 | +10.3 | +3.5 |

Table 1: Accuracy scores on geometry problems, graph algorithms, mathematical functions, and game. SKETCHPAD **yields large performance gains on most tasks and outperform all baselines.**

## 4.1 Results

We evaluate the performance of SKETCHPAD on multimodal LMs with API access, including `gpt-4-turbo-2024-04-29` and `gpt-4o-2024-05-13`. We compare these results to baselines without the Visual Sketchpad and other notable closed-source models, such as Claude 3 and Gemini-Pro, as well as open-source models like Mistral [19] and LLaMA-2 70B [46].

**Main results.** As shown in Table 1, SKETCHPAD consistently improves base model performance across all tasks, with an average improvement of 11.2% for `GPT-4o` and 23.4% for `GPT-4 Turbo`. In particular, we observe large gains on graph algorithms such as maximum flow and connectivity. For instance, `GPT-4o` with SKETCHPAD achieves an accuracy of 66.3% on the maximum flow problem, improving over the base model by 41.3%. Similarly, SKETCHPAD substantially improves the performance on mathematical functions, with `GPT-4 Turbo` achieving over 90% accuracy and `GPT-4o` over 88% accuracy on convexity and parity classification tasks. Furthermore, we observe gains (3% ∼ 10%) on game strategies, demonstrating that drawn game boards drawn can improve reasoning about game strategies. Overall, these results highlight the effectiveness of SKETCHPAD in enhancing the reasoning capabilities of multimodal language models across diverse domains.

## 5 Sketching to Solve Computer Vision Tasks

In this section, we experiment with SKETCHPAD on complex visual reasoning tasks. Recent work (BLINK) [9] finds that many core visual perception abilities are still missing from existing multimodal LMs—even though many computer vision specialist models possess such abilities. Also, SoM [55] shows that drawing segmentation masks on images unleashes the strong visual grounding ability of GPT-4V. We generalize these ideas with SKETCHPAD, allowing LMs to use **specialist vision models** to sketch. Details of these modules are in §5.1. SKETCHPAD enhances multimodal LMs' visual reasoning abilities and establishes new SOTAs on all 7 tasks (§5.2).

**Tasks.** We experiment with a wide range of complex visual reasoning tasks: (1) $V^*$**Bench** [51]. This benchmark contains questions about small items in an image. (2) **MMVP** benchmark from *Eyes Wide Shut* [44]. This benchmark contains visual questions specially designed to reveal the visual shortcomings of CLIP-based multimodal LMs. (3) **BLINK** [9]. This benchmark contains visual perception tasks that are easy for humans, but post significant challenge for multimodal LMs. Specifically, we experiment with relative depth, spatial reasoning, jigsaw puzzle, visual correspondence, and semantic correspondence tasks. More details of each task are in §D.

### 5.1 Vision Specialists as Sketching Tools in SKETCHPAD

LMs can use the following modules to sketch and manipulate images. We wrap these modules into Python functions that the LMs can call. Refer to §C for the function definitions.

**Detection.** This module takes an image and a simple text query (e.g., "cat") as input. We run the Grounding-DINO [30] open-vocabulary objection detection model and plot the detected bounding boxes (together with a number label) on the image. It also returns the bounding box coordinates.

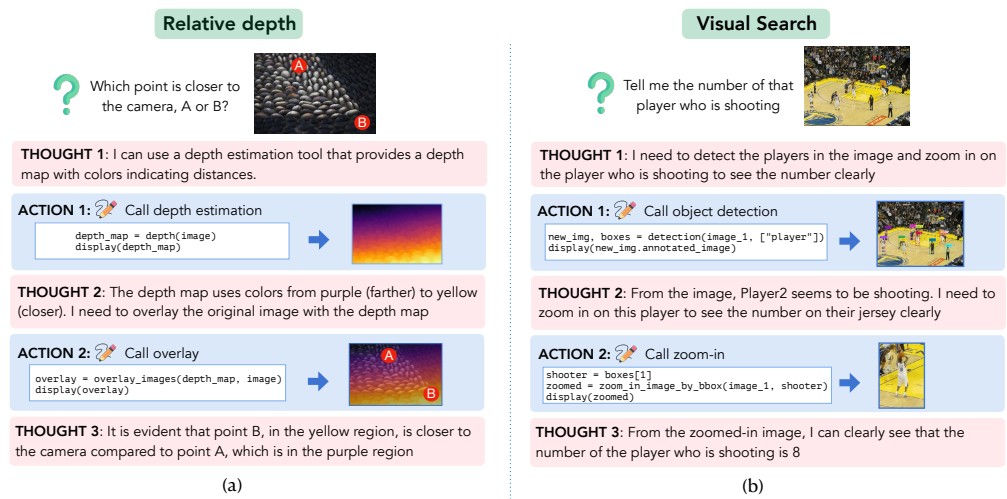

Figure 3: Examples of SKETCHPAD applied to vision tasks. The figure shows actual outputs generated by SKETCHPAD. By contrast, the baseline GPT-4o model cannot answer these questions correctly. Note that for demonstration purposes, the "A" and "B" marks in (a) are different from the actual images in the experiments.

**Segmentation.** This module takes an image as input and returns an image with colorful segmentation masks on it. Each mask also has a number label. We follow the implementation of SoM [55]. The underlying segmentation models are SegmentAnything [22] and Semantic-SAM [24].

**Depth estimation.** This module takes an image as input and returns a depth map. The underlying model is DepthAnything [56].

**Visual search via sliding window.** This module mimics how humans search for small items on an image. It takes a text query as input and runs a sliding window over the image. The window size is 1/3 of the image size, and the step size is 2/9 of the image size (so an image will have $4 \times 4 = 16$ windows). It returns a sequence of image patches in which the query is detected.

**Other image manipulation modules.** Other modules include (1) **zoom-in and crop**, which takes an image and a bounding box as input and returns the image patch inside the box; (2) **Overlay images**, which takes two images and alpha values as input, and returns the overlayed image.

## 5.2 Results

We experiment with the same multimodal LMs as in §4 on complex visual reasoning tasks. We compare the performance with and without SKETCHPAD, as well as other notable multimodal LMs, including Gemini [42], Claude 3 [2], and the open-source LLaVA 1.5 [26], LLaVA-NeXT [27].

**Main results.** Table 2 shows the performance of our SKETCHPAD and baselines. SKETCHPAD consistently improves base model performance across all tasks. GPT-4o with SKETCHPAD sets the new state-of-the-art results on all tasks. SKETCHPAD is particularly effective on $V^*$Bench, yielding 18.5% accuracy improvement for GPT-4 Turbo and 14.3% improvement for GPT-4o, surpassing the previous state of the art SEAL [51] which used a visual search model specifically trained for this task. On BLINK tasks, SKETCHPAD on average yields 6.6% absolute accuracy gain for GPT-4 Turbo and 9.0% gain for GPT-4o. Interestingly, despite the fact that all modules in SKETCHPAD work on a single image, the LMs also get substantial improvement on multi-image tasks, including jigsaw puzzles, visual correspondence, and semantic correspondence. Finally, GPT-4o, the LM with stronger multimodal ability than GPT-4 Turbo, benefits more from SKETCHPAD. For example, on the relative depth task, GPT-4o gets 12.1% accuracy improvement, while GPT-4 Turbo only gets 2.4%, showing that GPT-4o is better at understanding the depth map SKETCHPAD generated. Overall, our experiments show that SKETCHPAD is an effective way to improve multimodal LMs' performance on visual reasoning tasks.

**How many times is each vision specialist used?** We count the number of times each vision specialist is used in each task, as shown in Figure 4. Here we choose the four tasks that achieve the largest improvement: $V^*$Bench, relative depth, spatial reasoning, and semantic correspondence.

| Model | $V^*$Bench | MMVP | Depth | Spatial | Jigsaw | Vis. Corr. | Sem. Corr. |
|---|---|---|---|---|---|---|---|
| *Prior multimodal LLMs* | | | | | | | |
| LLaVA-1.5-7B [26] | 48.7 | - | 52.4 | 61.5 | 11.3 | 25.6 | 23.0 |
| LLaVA-1.5-13B [26] | - | 24.7 | 53.2 | 67.8 | 58.0 | 29.1 | 32.4 |
| LLaVA-NeXT-34B [27] | - | - | 67.7 | 74.8 | 54.7 | 30.8 | 23.7 |
| Claude 3 OPUS [2] | - | - | 47.6 | 58.0 | 32.7 | 36.6 | 25.2 |
| Gemini-Pro [42] | 48.2 | 40.7 | 40.3 | 74.8 | 57.3 | 42.4 | 26.6 |
| GPT-4V-preview [35] | 55.0 | 38.7 | 59.7 | 72.7 | 70.0 | 33.7 | 28.8 |
| Previous state of the art | 75.4 [51] | 49.3 [10] | 67.7 [27] | 76.2 [43] | 70.0 [35] | 42.4 [42] | 33.1 [49] |
| *Latest multimodal LLMs + Visual Sketchpad* | | | | | | | |
| GPT-4 Turbo | 52.5 | 71.0 | 66.1 | 68.5 | 64.7 | 48.8 | 30.9 |
| + Sketchpad | 71.0 | 73.3 | 68.5 | 80.4 | 68.5 | 52.3 | 42.4 |
| | +18.5 | +2.3 | +2.4 | +11.9 | +3.8 | +3.5 | +11.5 |
| GPT-4o | 66.0 | 85.3 | 71.8 | 72.0 | 64.0 | 73.3 | 48.6 |
| + Sketchpad | **80.3** | **86.3** | **83.9** | **81.1** | **70.7** | **80.8** | **58.3** |
| | +14.3 | +1.0 | +12.1 | +9.1 | +6.7 | +7.5 | +9.7 |

Table 2: Accuracy on complex visual reasoning tasks. SKETCHPAD **enhances both GPT-4 Turbo and GPT-4o performance, establishing new SOTA performance levels on all the tasks.**

We observe that (1) **the use of vision specialist is task-dependent, and the two LMs analyzed utilize similar tools.** For example, for $V^*$, which needs to locate small objects, the LMs mainly use detection, sliding window search, and zoom-in, similar to how people would search. For the relative depth task, both models rely on depth estimation. For spatial reasoning, the LMs use detection and segmentation to facilitate visual reasoning. (2) **GPT-4o likes to use more tools.** GPT-4o uses the vision specialists more often than GPT-4 Turbo. Also, the two LMs behave differently for the semantic correspondence tasks. GPT-4o uses the segmentation module for $40\%$ of the task instances, while GPT-4 Turbo uses the detection module for less than $20\%$ of times, and rarely uses the segmentation module. This difference may explain the performance gap between the two LMs ($58.3\%$ v.s. $42.4\%$) on this task.

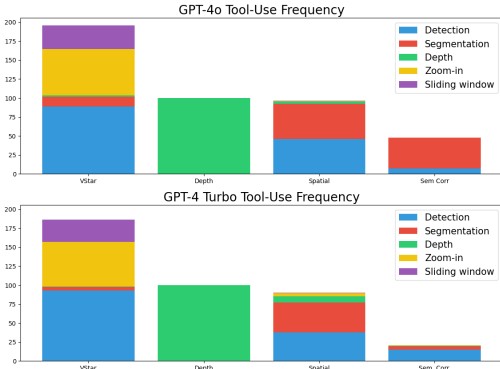

Figure 4: Percentage of times GPT-4o and GPT-4 Turbo use a visual module in SKETCHPAD when solving $V^*$Bench, relative depth, spatial reasoning, and semantic correspondence tasks.

| Model | $V^*$ | MMVP | Depth | Spatial |
|---|---|---|---|---|
| GPT-4 Turbo | 52.5 | 71.0 | 66.1 | 68.5 |
| SoM | 42.0 | 60.7 | 58.9 | 78.3 |
| SoM + orig. | 51.3 | **74.3** | 66.9 | 79.7 |
| Visprog | 33.2 | 16.3 | 67.8 | 53.8 |
| Sketchpad | **71.0** | 73.3 | **68.5** | **80.4** |
| GPT-4o | 66.0 | 85.3 | 71.8 | 72.0 |
| SoM | 49.0 | 70.7 | 62.9 | **83.2** |
| SoM + orig. | 68.1 | 84.0 | 75.0 | 82.5 |
| Visprog | 32.4 | 17.3 | 46.8 | 37.8 |
| Sketchpad | **80.3** | **86.3** | **83.9** | 81.1 |

Table 3: Comparison with other augmentation frameworks for multimodal LMs on single-image tasks. For fair comparison, we modify the original Visprog [14] framework by replacing the LM and VQA components with the corresponding GPT-4 model.

**Comparison with visual prompting and tool-use frameworks.** In Table 3, we compare SKETCH-PAD with the visual prompting framework **SoM** [55] and the LLM tool-use framework **Visprog** [14]. Details of these methods can be found in §2. For a fair comparison, we make the following adaptations: (1) we find that prompting LMs with SoM images can hurt performance, likely because the visual prompts confuse the model. To make a stronger baseline, we prompt the LM with both the original image and the SoM image (full prompt in §C), which we refer as "SoM + orig." (2) We replace the LM and VQA modules in Visprog with the corresponding GPT-4 model. (3) Since baseline methods are developed on single-image tasks, we compare SKETCHPAD on such tasks. From Table 3, we can see that **SKETCHPAD is the only framework that yields consistent improvement on all tasks.** SoM can boost spatial reasoning ability, as the authors reported. However, it can hurt the

performance on other tasks, even in the "SoM + orig." setting. Visprog performs worse than the base LM on all the tasks. As prior work [21, 18] suggests, one possible reason is that the vision modules themselves have errors, and the error propagates when the modules are composed by a program.

## 6 Analysis and Discussion

**Why does SKETCHPAD work?** First, **vision is a versatile and informational interface that complements language**. Dense information like depth and segmentation cannot be described easily through language [9]. In a broader perspective, humans have developed many visualization techniques that are direct, efficient, and informational. SKETCHPAD provides LMs the opportunity to use them. Second, in SKETCHPAD, multimodal LMs can **plan and reason based on the intermediate visual artifacts** they created. In contrast, in prior modular vision work [14, 38, 55], multimodal modules follow a predefined plan by either humans or code. SKETCHPAD is much more flexible and robust to errors. For example, suppose object detection makes an error. The LM can (in principle) find the error by viewing the bounding boxes, and change its following plans, but prior methods cannot. Third, as discussed next, **the plans of multimodal LMs are similar to human plans**, and therefore likely benefit from the fact that the underlying LMs have seen data with similar reasoning patterns.

**Do LMs have the same plans as humans?** We conduct a human study on all geometry problems and 10 problems on each vision task. On geometry, humans draw the same auxiliary line as GPT-4o 80% of the time. On vision, we show 2 human subjects the full plan of GPT-4o, which they rate is valid in 92.8% of instances. Most errors are caused by failures in the vision specialists (e.g., fail to detect an object) and mistakes in simple visual question answering, rather than planning.

| Model | Geometry | Maxflow | Convexity | Winner ID |
|---|---|---|---|---|
| LLaVA-NeXT-13B | 11.1 | 7.8 | 50.39 | 5.8 |
| + oracle Sketchpad | 22.2 | 10.2 | 50.0 | 36.7 |
| LLaVA-NeXT-34B | 26.1 | 0.8 | 81.6 | 49.0 |
| + oracle Sketchpad | 28.3 | 14.1 | 87.1 | 49.4 |

Table 4: Open-source LLaVA models' performance on math tasks. The oracle Sketchpad uses the visual artifact generated in the last action of GPT-4o + SKETCHPAD as inputs.

**Experiments on open-source models.** Can sketches like diagrams, plots, and auxiliary lines facilitate existing open-source multimodal LMs? To answer this question, we conduct the experiments in Table 4. We use the visual artifacts generated in the last action of GPT-4o + SKETCHPAD experiment as the image input for open-source LLaVA-NEXT models [27]. We can see that this oracle SKETCHPAD brings consistent improvement to math tasks and boosts mathematical reasoning.

## 7 Conclusion

We present Visual SKETCHPAD, a framework that provides multimodal LMs with the tools necessary to generate intermediate sketches to reason over tasks. For complex mathematical reasoning tasks, SKETCHPAD yields large performance gains, by visualizing auxiliary lines, math functions, graphs, and games during reasoning. For visual reasoning tasks, we add vision specialists to SKETCHPAD. The LM can call these specialists during reasoning, observing the visualization of these specialists' predictions (e.g., bounding boxes from the object detection model; masks from the segmentation model), and then conduct further planning and reasoning. Experiments show that SKETCHPAD enhances the LMs' performance across all tasks, and sets new state-of-the-art results. Ultimately, SKETCHPAD represents a step toward endowing LMs with more human-like multimodal intelligence, leveraging the complementary strengths of language and vision to tackle increasingly complex reasoning challenges.

**Limitations and future directions.** First, SKETCHPAD requires more computing resources than directly outputting language tokens. We discuss more about computing costs in E. Second, this work focuses on existing off-the-shelf LMs. Future work may explore the training side of SKETCHPAD. For example, recent multimodal models like Unified-IO 2 [31] and Chameleon [41] are natively multimodal and can output both text and images. SKETCHPAD may emerge as a new paradigm for instruction tuning these models. Finally, SKETCHPAD can be applied in more areas. For example, for robotics, we can apply SKETCHPAD to search for small things in a crowded space, highlight the object of interest, and zoom the camera for a better view or use depth estimation to help navigation.

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

# Appendices

# A  Acknowledgement

This work is in part supported by gifts from Allen Institute for Artificial Intelligence, Meta, Apple, and Amazon Science. We also greatly appreciate the help from Zixian Ma, Wei-Chiu Ma, Benlin Liu, Jieyu Zhang, UW NLP, AllenNLP, AI2 PRIOR, and UW RAIVN for their help.

# B  More Qualitative Examples

Here we provide more details of how GPT-4o and SKETCHPAD solve complex mathematical and visual reasoning tasks.

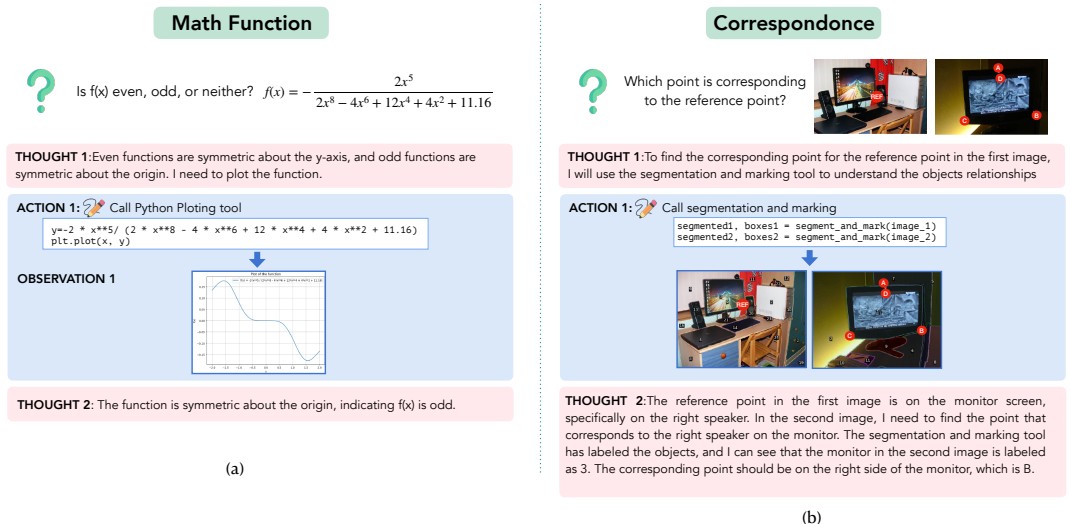

Figure 5: **Qualitative examples** of how GPT-4o and SKETCHPAD solve (a) complex mathematical and (b) visual reasoning tasks.

# C  Prompts

**Prompts for computer vision tasks.** Here we provide the running log of GPT-4o + SKETCHPAD tackling a visual search problem from $V^*$Bench [51]. We use the same prompt template for all computer vision problems in this work. For visualization purposes, we present the prompts that contain codes differently. In our implementation, they are all text inputs to the LLM.

---

**SYSTEM PROMPT**
You are a helpful multimodal AI assistant. [MORE INSTRUCTIONS ...]

For each turn, you should first do a "THOUGHT", based on the images and text you see. If you think you get the answer to the intial user request, you can reply with "ANSWER: <your answer>" and ends with "TERMINATE".

---

**Initial Prompt + Request**

---

```
1   Here are some tools that can help you. All are python codes. They are in tools .py and will be imported for you.
2   The images has their own coordinate system. The upper left corner of the image is the origin (0, 0). All coordinates are normalized, i.e., the
    range is [0, 1].
3   All bounding boxes are in the format of [x, y, w, h], which is a python list. x is the horizontal coordinate of the upper−left corner of the box, y
    is the vertical coordinate of that corner, w is the box width, and h is the box height.
4   Notice that you, as an AI assistant, is not good at locating things and describe them with coordinate. You can use tools to generate bounding boxes
    .
5   You are also not good at answering questions about small visual details in the image. You can use tools to zoom in on the image to see the details.
    Below are the tools in tools .py:
6   ``` python
7   class AnnotatedImage:
8       # A class to represent an annotated image. It contains the annotated image and the original image.
9
10      def __init__ ( self , annotated_image: Image.Image, original_image : Image.Image=None):
11          self .annotated_image = annotated_image
12          self . original_image  = original_image
13
14  def detection (image, objects ):
15      """Object detection using Grounding DINO model. It returns the annotated image and the bounding boxes of the detected objects .
16      The text can be simple noun, or simple phrase (e.g., 'bus', 'red car'). Cannot be too hard or the model will break.
17      The detector is not perfect , it may wrongly detect objects or miss some objects.
18      Also, notice that the bounding box label might be out of the image boundary.
19      You should use the output as a reference , not as a ground truth .
20      When answering questions about the image, you should double−check the detected objects .
21
22      Args:
23          image (PIL.Image.Image): the input image
24          objects  (List [ str ]): a list of objects to detect . Each object should be a simple noun or a simple phrase. Should not be hard or abstract
    concepts like "text" or "number".
25
26      Returns:
27          output_image (AnnotatedImage): the original image, annotated with bounding boxes. Each box is labeled with the detected object , and an
    index.
28          processed boxes (List ): listthe bounding boxes of the detected objects
29
30      Example:
31          image = Image.open("sample_img.jpg")
32          output_image, boxes = detection (image, ["bus"])
33          display (output_image.annotated_image)
34          print (boxes) # [[0.24,  0.21,  0.3,  0.4], [0.6,  0.3,  0.2,  0.3]]
35          # you need to double−check the detected objects . Some objects may be missed or wrongly detected .
36      """
37
38  def sliding_window_detection (image, objects ):
39      """Use this when you are searching for objects in the image, but the objects are not detected by the object detection model.
40      In that case, the most common reason is that the object is too small such that both the vision−language model and the object detection model
    fail to detect it .
41      This function tries to detect the object by sliding window search.
42      With the help of the detection model, it tries to detect the object in the zoomed−in patches.
43      The function returns a list of annotated images that may contain at leas one of the objects , annotated with bounding boxes.
44      It also returns a list of a list of bounding boxes of the detected objects .
45
46      Args:
47          image (PIL.Image.Image): the input image
48          objects  (List [ str ]): a list of objects to detect . Each object should be a simple noun or a simple phrase. Should not be hard or abstract
    concepts like "text" or "number".
49
50      Returns:
51          possible_patches  (List [AnnotatedImage]): a list of annotated zoomed−in images that may contain the object , annotated with bounding boxes.
52          possible_boxes  (List [ List [ List [ Float ]]]) : For each image in possible_patches , a list of bounding boxes of the detected objects .
53              The coordinates are w.r.t. each zoomed−in image. The order of the boxes is the same as the order of the images in possible_patches .
54
55      Example:
56          image = Image.open("sample_img.jpg")
```

```python
57              possible_patches ,  possible_boxes  = search_object_and_zoom(image, [" bird ",  " sign "])
58              for  i ,  patch  in  enumerate( possible_patches ):
59                  print (f"Patch {i }:")
60                  display (patch .annotated_image)
61
62              #  print  the  bounding  boxes  of  the  detected  objects  in  the   first  patch
63              print ( possible_boxes [0]) #  [[0.24,  0.21,  0.3,  0.4],  [0.6,  0.3,  0.2,  0.3]]
64          """
65
66  def  segment_and_mark(image, anno_mode:list = ['Mask', 'Mark']):
67          """Use a  segmentation  model to  segment  the  image, and  add  colorful  masks on  the  segmented objects . Each segment  is  also  labeled  with a  number.
68          The annotated  image  is  returned  along  with  the  bounding  boxes  of  the  segmented  objects .
69          This tool  may  help  you to  better  reason  about  the  relationship  between objects , which can  be  useful  for  spatial  reasoning  etc .
70          DO  NOT  use  this  tool  to  search  or  detect  an  object .  It  is  likely  the  object  is  small  and segmentaiton  does  not  help .
71          Segmentation  and  marking  can  also  be  helpful  for  3D and  video  reasoning . For  example, helping  you  to  see  more  clearly  and  analyzes  the
        relationship  between  different  frames  of  a  video .
72
73          Args:
74              image (PIL.Image.Image): the  input  image
75              anno_mode (list ,  optional ):  What annotation  is  added  on  the  input  image. Mask is  the  colorful  masks. And mark is  the  number labels .
        Defaults  to  ['Mask', 'Mark'].
76
77          Returns:
78              output_image  (AnnotatedImage):  the  original  image annotated  with  colorful  masks and number labels . Each mask  is  labeled  with  a  number. The
        number label  starts   at  1.
79              bboxes (List ):  listthe  bounding boxes  of  the  masks.The order  of  the  boxes  is  the  same as  the  order  of  the  number labels .
80
81          Example:
82              User request :  I  want  to  find  a  seat  close  to  windows, where should  I  sit ?
83              Code:
84              ``` python
85              image = Image.open("sample_img.jpg")
86              output_image,  bboxes  = segment_and_mark(image)
87              display (output_image.annotated_image)
88              ```
89              Model reply:  You can  sit  on  the  chair  numbered as  5,  which  is  close  to  the  window.
90              User:  Give me  the  bounding box  of  that  chair .
91              Code:
92              ``` python
93              print (bboxes[4])  #  [0.24,  0.21,  0.3,  0.4]
94              ```
95              Model reply:  The bounding box  of  the  chair  numbered as  5  is  [0.24,  0.21,  0.3,  0.4].
96          """
97
98  def  depth(image):
99          """Depth  estimation  using DepthAnything model. It  returns  the  depth  map of  the  input  image.
100         A colormap  is  used  to  represent  the  depth . It  uses Inferno  colormap. The closer  the  object , the  warmer the  color .
101         This tool  may help  you  to  better  reason  about  the  spatial   relationship , like  which object  is  closer  to  the  camera.
102
103         Args:
104             image (PIL.Image.Image):  the  input  image
105
106         Returns:
107             output_image  (PIL.Image.Image): the  depth  map of  the  input  image
108
109         Example:
110             image = Image.open("sample_img.jpg")
111             output_image = depth(image)
112             display (output_image)
113         """
114
115 def  zoom_in_image_by_bbox(image, box, padding=0.05):
116         """A  simple  wrapper  function  to  crop  the  image based  on  the  bounding box.
117         When you want  to  answer question  about  visual   details  in  a  bounding box annotated  by  the  detection  tool , you  would like  to  zoom  in  on  the
        object  using  this  function .
118
119         Args:
120             image (PIL.Image.Image): the  input  image
121             box  (List [ float ]):  the  bounding box  in  the  format  of  [x,  y,  w, h]
122             padding ( float ,  optional ): The padding  for  the  image crop ,  outside  of  the  bounding box. Defaults  to  0.1. The zoom factor  cannot  be  too
        small .  Minimum is 0.05
123
124         Returns:
125             cropped_img  (PIL.Image.Image): the  cropped image
126
127         Example:
128             image = Image.open("sample_img.jpg")
129             annotated_img,  boxes  = detection (image, "bus")
130             cropped_img = zoom_in_image_by_bbox(image, boxes[0], padding=0.05)
131             display (cropped_img)
132         """
133
134 def  overlay_images(background_img, overlay_img, alpha=0.3,  bounding_box=[0, 0,  1,  1]):
135         """
136         Overlay  an  image onto  another  image with  transparency .
137         This  is  particularly   useful  visualizing  heatmap while  preserving  some info  from  the  original  image.
138         For  example, you can  overlay  a  segmented image on a  heatmap to  better  understand  the  spatial   relationship  between objects .
139         It  will  also  help  seeing  the  labels ,  circles  on  the  original  image that  may not  be  visible  on  the  heatmap.
140
141         Args:
142         background_img_pil (PIL.Image.Image): The background image in  PIL format .
143         overlay_img_pil  (PIL.Image.Image): The image to  overlay  in  PIL format .
```

```
144        alpha ( float ): Transparency of the overlay image.
145        bounding_box (List[ float ]): The bounding box of the overlay image. The format is [x, y, w, h]. The coordinates are normalized to the
     background image. Defaults to [0, 0, 1, 1].
146
147        Returns:
148        PIL.Image.Image: The resulting image after overlay, in PIL format.
149        s
150        Example:
151            image = Image.open(' original .jpg')
152            depth_map = depth(image)
153            overlayed_image = overlay_images(depth_map, image, alpha=0.3)
154            display (overlayed_image)
155        """
156  ```
157  # GOAL #: Based on the above tools , I want you to reason about how to solve the # USER REQUEST # and generate the actions step by step (each action
     is a python jupyter notebook code block) to solve the request .
158  You may need to use the tools above to process the images and make decisions based on the visual outputs of the previous code blocks .
159  Your visual ability is not perfect, so you should use these tools to assist you in reasoning about the images.
160
161  # [ six in−context examples here ]
162
163  # USER REQUEST #:
```

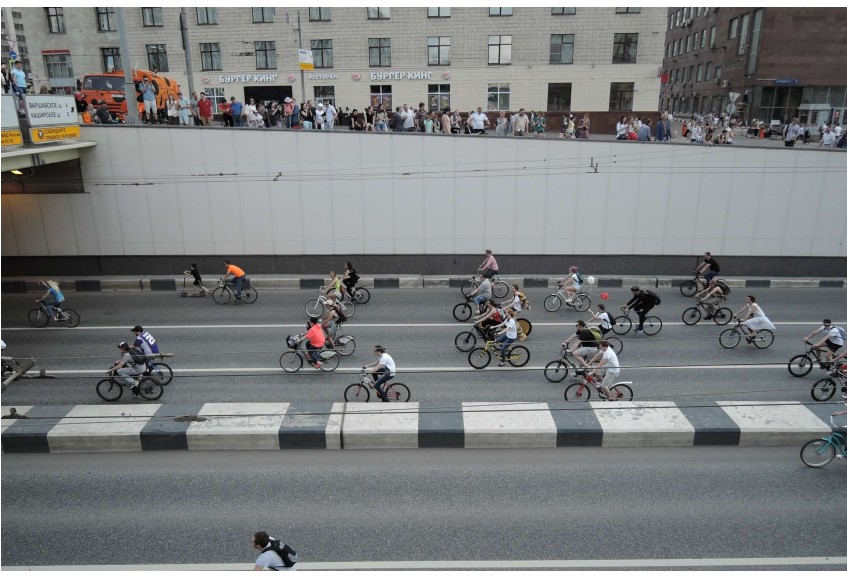

```
1   Is the red balloon above or below the white balloon? Options: (A) The red balloon is below the white balloon . (B) The red balloon is above the
    white balloon .
2
3   # USER IMAGE stored in image_1 as PIL image.
4   Now please generate only THOUGHT 0 and ACTION 0 in RESULT. If no action needed, also reply with ANSWER: <your answer> and ends with TERMINATE
    in the RESULT:
5
6   # RESULT #:
```

## Thought + Action 0

```
1   THOUGHT 0: To determine the relative positions of the red and white balloons , I need to detect the balloons in the image. I will use the detection
    tool to find the balloons .
2   ACTION 0:
3   ``` python
4   image_1_balloon_detection , image_1_balloon_boxes = detection (image_1, ["red balloon", "white balloon"])
5   display (image_1_balloon_detection .annotated_image)
6   ```
```

---
**Observation 0**

---

Execution success. The output is as follows.

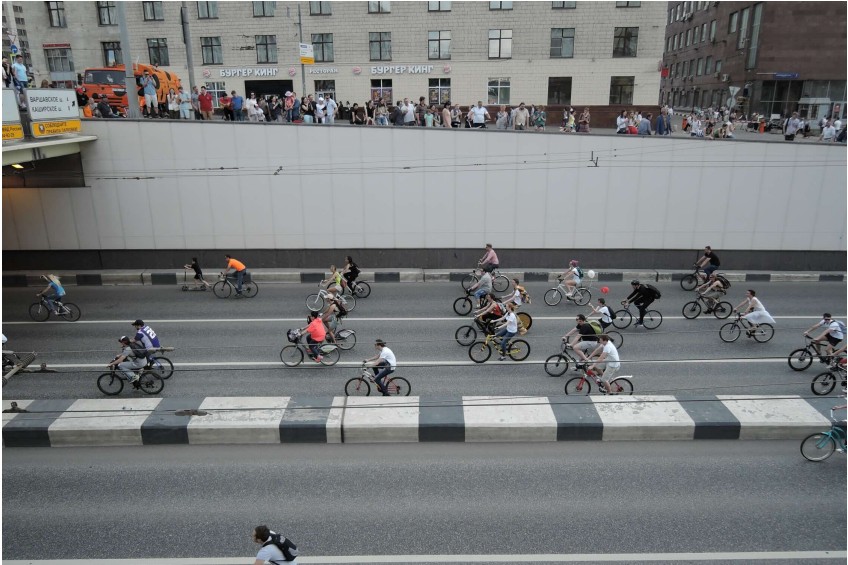

Please generate the next THOUGHT and ACTION. If you can get the answer, please also reply with
ANSWER: <your answer> and ends with TERMINATE.

---
**Thought + Action 1**

---

```
1  THOUGHT 1: The detection tool did not annotate the balloons in the image. I will use the sliding_window_detection tool to search for the red and
   white balloons in the image.
2
3  ACTION 1:
4  ``` python
5  possible_patches , possible_boxes = sliding_window_detection(image_1, ["red balloon", "white balloon"])
6  for i, patch in enumerate( possible_patches ):
7      print (f"Patch {i+1}:")
8      display (patch.annotated_image)
9  ```
```

---
**Observation 1**

---

Execution success. The output is as follows. Patch 1:

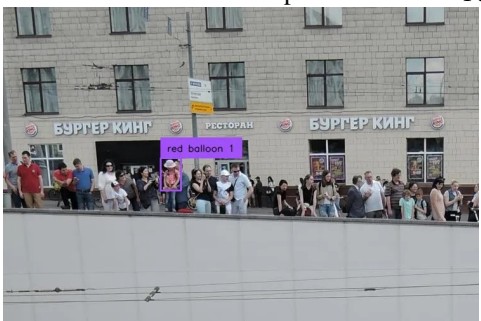

Patch 2:

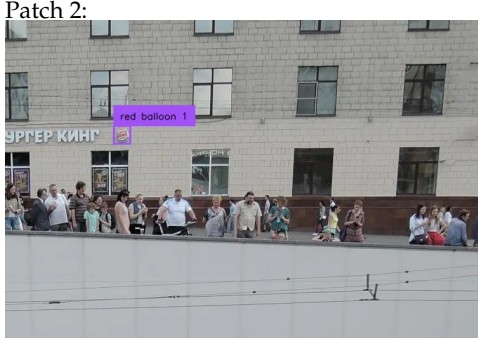

Patch 3:

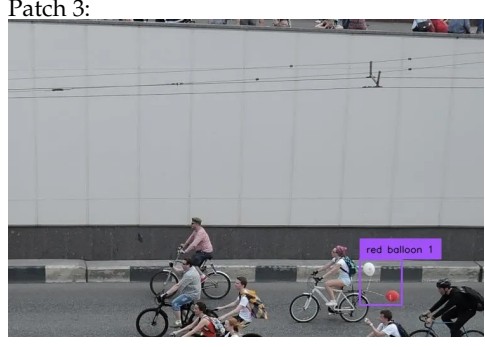

Patch 4:

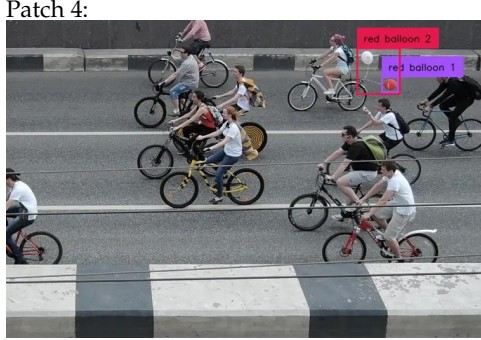

Patch 5:

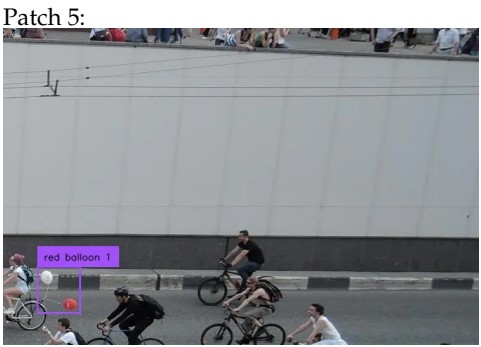

Please generate the next THOUGHT and ACTION. If you can get the answer, please also reply with ANSWER: <your answer> and ends with TERMINATE.

---

**Thought + Action 2**

---

THOUGHT 2: From Patch 4 and 5, I can see both the red and white balloons. The red balloon is clearly below the white balloon in these patches.
ACTION 2: No action needed.
ANSWER: The red balloon is below the white balloon. So the answer is (A). TERMINATE

**Prompts for math tasks.** The prompts are similar to the vision task for math tasks, except we remove the computer vision specialists, and add example codes for math plotting. Besides, the user query is different. For each task, the user query part of the prompt is as follows.

---

**PROMPT**

You are given a real-valued, scalar function $f(x)$.
YOUR TASK is to determine whether $f(x)$ is an even function, an odd function, or neither. Definition of an odd function: A function such that

$$f(-x) = -f(x)$$

where the sign is reversed but the absolute value remains the same if the sign of the independent variable is reversed. A function is neither even nor odd if it does not satisfy either condition.
Here is the expression of $f(x)$:

$$f(x) = \frac{-2x^5}{2x^8 - 4x^6 + 12x^4 + 4x^2 + 11.16}$$

Respond with 'even', 'odd', 'neither' first on whether the function $f(x)$ is even, odd, or neither, based on the definitions and your observation of the function. You can generate matplotlib code to visualize the function.
If you can get the answer, please reply with ANSWER: <your answer>, extract the final answer in FINAL ANSWER: <final answer> and ends with TERMINATE in the RESULT.
*Answer:*

---

Figure 6: Prompt for the Math Parity task. We follow a similar prompt format to [8], except prompting the models to write the code to generate images.

---

**PROMPT**

You are given a real-valued, scalar function $f(x)$.
YOUR TASK is to determine whether $f(x)$ is an convex function or concave function. Definition of a convex function: A function such that

$$\forall x, y, 0 \le t \le 1, f(tx + (1-t)y) \le tf(x) + (1-t)f(y)$$

Definition of a concave function: A function such that

$$\forall x, y, 0 \le t \le 1, f(tx + (1-t)y) \ge tf(x) + (1-t)f(y)$$

Here is the expression of $f(x)$:
$$f(x) = 7.57 - 0.08 * Abs(x)$$

Respond with 'convex' or 'concave' first on whether the function f (x) is convex or concave, based on the definitions and your observation of the function. You can generate matplotlib code to visualize the function.
If you can get the answer, please reply with ANSWER: <your answer>, extract the final answer in FINAL ANSWER: <final answer> and ends with TERMINATE in the RESULT.
*Answer:*

---

Figure 7: Prompt for the Math Convexity task. We follow the similar prompt format to [8], except prompting the models to write the code to generate images.

**PROMPT**

You are given an adjacency matrix of a graph and two query nodes.

YOUR TASK is to find if there is a path between the two nodes.

*Definition of connectivity:* In an undirected graph G, two vertices u and v are called connected if G contains a path from u to v. A path in a graph is a finite sequence of edges which joins a sequence of vertices. In the query example, the nodes and the adjacency matrix are zero-indexed.

Query Example:

*Adjacency Matrix:*

$$
\begin{bmatrix}
0 & 0 & 0 & 0 & 0 & 1 & 0 & 0 & 0 \\
0 & 0 & 1 & 0 & 0 & 0 & 1 & 0 & 0 \\
0 & 1 & 0 & 0 & 1 & 0 & 0 & 0 & 0 \\
0 & 0 & 0 & 0 & 0 & 0 & 0 & 0 & 0 \\
0 & 0 & 1 & 0 & 0 & 0 & 0 & 0 & 0 \\
1 & 0 & 0 & 0 & 0 & 0 & 1 & 0 & 1 \\
0 & 1 & 0 & 0 & 0 & 1 & 0 & 0 & 0 \\
0 & 0 & 0 & 0 & 0 & 0 & 0 & 0 & 0 \\
0 & 0 & 0 & 0 & 0 & 1 & 0 & 0 & 0
\end{bmatrix}
$$

*Query nodes indices (zero-indexed)*: 4 and 0

Respond with 'yes' or 'no' first on whether the query nodes are connected or not in the graph.

If there is a path, first provide the path as a sequence of vertices (nodes), and then explain your reasoning. You can use networkx to draw the graph. If there is no path, explain why in details. Answer (start with 'yes' or 'no'): If you can get the answer, please reply with ANSWER: <your answer>, extract the final answer in FINAL ANSWER: <final answer> and ends with TERMINATE in the RESULT.

*Answer:*

Figure 8: Prompt for the Graph Connectivity task. We follow the similar prompt format to [8], except prompting the models to write the code to generate images.

**PROMPT**
You are given a visual representation of two graphs, graph G on the left and graph H on the right. YOUR TASK is to determine whether the two graphs are isomorphic to each other.
*Definition of graph isomorphism:* In graph theory, an isomorphism of graphs G and H is a bijection $f$ between the vertex sets of G and H, denoted as $f : V(G) \rightarrow V(H)$. G and H are said to be isomorphic when $f$ satisfies the following: any two vertices $u$ and $v$ of G are adjacent in G if and only if $f(u)$ and $f(v)$ are adjacent in H. This kind of bijection is commonly described as "edge-preserving bijection", in accordance with the general notion of isomorphism being a structure-preserving bijection.
In the query example, the adjacency matrices are zero-indexed.
*Adjacency Matrix G:*

$$
\begin{bmatrix}
0 & 0 & 0 & 0 & 0 & 1 & 0 & 0 & 0 \\
0 & 0 & 1 & 0 & 0 & 0 & 1 & 0 & 0 \\
0 & 1 & 0 & 0 & 1 & 0 & 0 & 0 & 0 \\
0 & 0 & 0 & 0 & 0 & 0 & 0 & 0 & 0 \\
0 & 0 & 1 & 0 & 0 & 0 & 0 & 0 & 0 \\
1 & 0 & 0 & 0 & 0 & 0 & 1 & 0 & 1 \\
0 & 1 & 0 & 0 & 0 & 1 & 0 & 0 & 0 \\
0 & 0 & 0 & 0 & 0 & 0 & 0 & 0 & 0 \\
0 & 0 & 0 & 0 & 0 & 1 & 0 & 0 & 0
\end{bmatrix}
$$

*Adjacency Matrix H:*

$$
\begin{bmatrix}
0 & 0 & 0 & 0 & 0 & 0 & 0 & 0 & 0 \\
0 & 0 & 1 & 0 & 0 & 0 & 1 & 0 & 0 \\
1 & 0 & 0 & 0 & 0 & 0 & 0 & 0 & 0 \\
0 & 0 & 0 & 0 & 0 & 0 & 0 & 0 & 0 \\
0 & 0 & 0 & 0 & 0 & 0 & 0 & 0 & 0 \\
1 & 0 & 0 & 0 & 0 & 0 & 1 & 0 & 0 \\
0 & 1 & 0 & 1 & 0 & 1 & 0 & 0 & 0 \\
0 & 0 & 0 & 0 & 0 & 0 & 0 & 1 & 0 \\
0 & 0 & 0 & 0 & 0 & 1 & 0 & 0 & 0
\end{bmatrix}
$$

Respond with 'yes' or 'no' first on whether the two graphs are isomorphic to each other. You can use networkx to draw the graph. If they are isomorphic, first provide the bijection between the two graphs, and then explain your reasoning. You can use networkx to draw the graph. If they are not isomorphic, explain why in detail. Answer (start with 'yes' or 'no'): If you can get the answer, please reply with ANSWER: <your answer>, extract the final answer in FINAL ANSWER: <final answer> and ends with TERMINATE in the RESULT.
*Answer:*

Figure 9: Prompt for the Graph Isomorphism task. We follow a similar prompt format to [8], except prompting the models to write the code to generate images.

**PROMPT**

You are given an adjacency matrix of a graph and two query nodes (one source node and one sink node). The source node is the node where the flow starts and the sink node is the node where the flow ends.

YOUR TASK is to solve the maxflow problem given the weighted directed graph.

*Definition of Maxflow problem:* In the max flow problem, we have a directed graph with a source node $s$ and a sink node $t$, and each edge has a capacity (integer valued, colored in green) that represents the maximum amount of flow that can be sent through it. The goal is to find the maximum amount of flow that can be sent from $s$ to $t$, while respecting the capacity constraints on the edges.

Query Example:

*Adjacency Matrix:*

$$\begin{bmatrix} 0 & 1 & 4 \\ 0 & 0 & 6 \\ 0 & 0 & 0 \end{bmatrix}$$

*Source node (zero-indexed)*: 0
*Sink node (zero-indexed)*: 2

In the query example, the nodes and the adjacency matrix are zero-indexed. You can use networkx to draw the graph. If you can get the answer, please reply with AN-SWER: <your answer>, extract the final answer in FINAL ANSWER: <final answer> and ends with TERMINATE in the RESULT.

*Answer:*

Figure 10: Prompt for Graph Maxflow task. We follow the similar prompt format to [8], except prompting the models to solve the maxflow problem.

**PROMPT**

Given the following FEN of the chess game:

1r1q1rk1/1b2b1Qp/4pp1B/pp1nP3/2pPN3/P1P5/1PB3PP/R4RK1 b - - 0 18

Determine the game's outcome. Who won: White or Black? Answer (start with 'white' or 'black' or 'draw'):

You can draw the chess board using Python given the FEN string. If you can get the answer, please reply with ANSWER: <your answer>, extract the final answer in FINAL ANSWER: <final answer> and ends with TERMINATE in the RESULT.

*Answer:*

Figure 11: Prompt for Winner ID task. We follow a similar prompt format to [8], except prompting the models to analyze the game outcome.

# D   Dataset Statistics

Table 5 and  6 show the statistics of the datasets we used, including IsoBench [8], BLINK [9], MMVP [44], and $V^*$Bench [51].

| Dataset | size | partition | representation |
|---|---|---|---|
| Math Parity | 383 | val | code |
| Math Convexity | 255 | val | code |
| Graph Maxflow | 128 | val | array |
| Graph Connectivity | 128 | val | array |
| Graph Isomorphism | 128 | val | array |
| Winner ID | 257 | val | FEN |

Table 5: IsoBench [8] data statistics.

| Dataset | size | partition | input |
|---|---|---|---|
| $V*$Bench | 257 | - | Single Image |
| MMVP | 300 | - | Single Image |
| BLINK Relative Depth | 124 | val | Single Image |
| BLINK Spatial Relation | 143 | val | Single Image |
| BLINK Jigsaw Puzzle | 150 | val | Multiple Images |
| BLINK Visual Correspondence | 172 | val | Multiple Images |
| BLINK Semantic Correspondence | 139 | val | Multiple Image |

Table 6: Vision tasks data statistics.

# E   Costs

The cost of running each task using GPT-4o is in Table 7.

| Dataset | tokens per sample | GPT-4o cost per sample |
|---|---|---|
| Math Parity | 2994 | $0.015 |
| Math Convexity | 2211 | $0.011 |
| Graph Connectivity | 2819 | $0.014 |
| Graph Isomorphism | 3143 | $0.016 |
| $V^*$Bench | 26647 | $0.133 |
| MMVP | 11870 | $0.059 |
| BLINK Relative Detph | 14078 | $0.070 |
| BLINK Spatial Relation | 12848 | $0.064 |
| BLINK Jigsaw Puzzle | 13206 | $0.066 |
| BLINK Visual Correspondence | 16988 | $0.085 |
| BLINK Semantic Correspondence | 11508 | $0.058 |

Table 7: The cost of running SKETCHPAD on each task.

# F   Impact Statement

Our work proposes SKETCHPAD, a framework aiming at advancing academic research and meeting industry needs. In a broader perspective, SKETCHPAD proposes a new way that humans can interact with LMs, and makes LMs more interpretable by eliciting their reasoning process with both language and sketches. On the other hand, if misused, the LMs may be used to generate harmful vision and text artifacts. Nevertheless, this is not directly related to our research, and more researchers can be involved to research on the safety issue in a multimodal context.

