# OpenReview forum: "Visual Sketchpad: Sketching as a Visual Chain of Thought for Multimodal Language Models"
_NeurIPS.cc/2024/Conference — NeurIPS 2024 poster_

### Official Review · Reviewer_jSat · 2024-07-09

**Soundness:** 3
**Presentation:** 3
**Contribution:** 3
**Rating:** 5
**Confidence:** 4

**Summary:**

This paper proposes visual sketchpad that aims to aid the existing multimodal language models (MLMs). Specifically, the visual sketchpad serves as a task-specific prompting technique that calls tools to draw sketches for the input problem as additional context to help solve the problem. This prompt technique is like a chain of thought but in a visual way, and the tools include code generation as image drawer as well as vision specialist models. Benchmarks show that the proposed visual sketchpad frameworks can significantly boost the performance of base models (e.g., GPT-4o and LLAVA-Next) on multiple math and vision tasks.

**Strengths:**

1. This paper proposes an interesting and effective framework (i.e., visual sketchpad) to assist the MLMs through chain of thought. Compared with text-only chain-of-thought prompts with external tools, the proposed visual sketchpad further expands the thought generation in a visual fashion, which is particularly suitable for certain scenarios like the geometry-based reasoning for math problems.

2. The authors design multiple tools that enables visual sketchpad for various tasks: code generation to draw diagram/figure for math problems, and visual specialists (including depth generation / sliding window / zoom-in / crop / overlay images) for visual reasoning tasks. The rich tools provide multiple combinations to analyze the input problem and generate suitable sketches for problem solving.

3. The proposed visual sketchpad does not need any training or finetuning, making it easy to integrate with existing MLMs. In the evalution, the GPT-4o / GPT-4 Turbo w/ visual sketchpad got significant improvement on geometry, graph, math, and game problems. In addition, the sketchpad also brings decent performance gain and achieves SOTA on several complex visual reasoning tasks.

4. The paper is well written and easy to comprehend.

**Weaknesses:**

1. As the proposed visual sketchpad attempts to generate code to draw the images. I wonder if it is possible that sometimes code has issues and cannot generate images properly. If so, how to deal with such issues.

2. For the geometry problem, can the visual sketchpad support geometry image (e.g., in jpg) as input? If so, how does the visual sketchpad overlay lines on the image?

3. Currently, the visual sketchpad is a set of different prompts that tackle specific problems with a corresponding set of tools. Can the visual sketchpad be a single prompt to call tools and tackle all the problems listed in the evaluation? If so, that would be useful as the prompt can be integrated into the system prompt for MLMs as an enhanced version.

4. The evaluation misses an important set of comparisions: There is no comparison between the proposed approach and the existing chain-of-thought frameworks.

5. Although most benchmarks in the evaluation show that the additional context from visual sketchpad brings benefits, I wonder if there are some cases that the additional context introduces noises and causes errors compared with the baseline model.

6. Typos
Line 64: jigwaw -> jigsaw

**Questions:**

Please see Weaknesses section for the questions.

**Limitations:**

Yes, the author discusses some limitations in Section 7.

---

> ### Author Rebuttal · Authors · 2024-08-06
>
> Thank you for your insightful and constructive review! We are honored that you believe visual sketchpad is interesting and effective. We address each question as follows. Hope that our response clarified your concerns, and we would be grateful if you could consider improving the rating after seeing our responses!
>
> **1. As the proposed visual sketchpad attempts to generate code to draw the images. I wonder if it is possible that sometimes code has issues and cannot generate images properly. If so, how to deal with such issues.**
>
> **Answer:** We thank the reviewer for pointing this out. Sometimes, LMs generate incorrect code that results in execution errors. Our framework addresses this problem by feeding the error message back to the model and prompting it to generate a revised version of the code. This idea was developed by a long line of prior work, for example, self-debug [1] and AutoGen[51]. We will add more discussions in the final version.
>
> [1] Chen, Xinyun, Maxwell Lin, Nathanael Schärli, and Denny Zhou. "Teaching large language models to self-debug." arXiv preprint arXiv:2304.05128 (2023).
>
>
> **2. For the geometry problem, can the visual sketchpad support geometry image (e.g., in jpg) as input? If so, how does the visual sketchpad overlay lines on the image?**
>
> **Answer:** This is a great question. We experimented with many multimodal LLMs (e.g., GPT-4o) and diffusion-based image editing models (e.g., SDXL-edit), but found that these models were unable to accurately add the required line to the geometry image. As a result, our current framework does not support using only the geometry image as input. Instead, we provide the matplotlib code for the geometry diagrams, enabling the LLM to add a line programmatically. Future work would be to develop or fine-tune an image-editing model specifically for overlaying precise lines on geometric images. We appreciate your question and will include this discussion in our future work section.
>
> **3. Currently, the visual sketchpad is a set of different prompts that tackle specific problems with a corresponding set of tools. Can the visual sketchpad be a single prompt to call tools and tackle all the problems listed in the evaluation? If so, that would be useful as the prompt can be integrated into the system prompt for MLMs as an enhanced version.**
>
> **Answer:** Thanks for your suggestion! Visual sketchpad uses a single unified prompt for all vision tasks and similarly, a single unified prompt for all math tasks. These prompts can be directly integrated into the system prompt of any MLMs.
>
> **4. The evaluation misses an important set of comparisions: There is no comparison between the proposed approach and the existing chain-of-thought frameworks.**
>
> **Answer:** Thanks for your feedback. We'd like to clarify that our GPT-4-turbo and GPT-4 baselines already incorporate chain-of-thought reasoning. These models produce multiple reasoning steps before outputting a final answer. It is widely believed that OpenAI trains their models to output chain-of-thought reasoning during the instruction-tuning and alignment processes. Here is a sample output from the baseline methods. As you can see, the baseline already employs a chain-of-thought approach:
>
> **Query:** You are given an adjacency matrix of a graph and two query nodes. \n\nYOUR TASK is to find if there is a path between the two nodes.\n\n    Example:\nAdjacency Matrix: [[0, 0, 0, 0, 0, 0, 0, 0, 0, 0, 0, 0], [0, 0, 0, 0, 0, 0, 0, 0, 0, 0, 0, 0], [0, 0, 0, 0, 0, 0, 0, 0, 0, 0, 0, 0], [0, 0, 0, 0, 0, 0, 0, 0, 0, 0, 0, 0], [0, 0, 0, 0, 0, 0, 0, 0, 0, 0, 0, 0], [0, 0, 0, 0, 0, 0, 0, 0, 0, 0, 0, 0]....
>
> **Baseline GPT-4 output:** The adjacency matrix provided represents an undirected graph where each row and column corresponds to a node, and the value at a given position (i, j) indicates whether there is a direct edge between node i and node j. A value of 1 represents an edge, and a value of 0 represents no edge.\n\nFrom the adjacency matrix, we can see that most nodes have no edges connecting them to any other nodes, except for nodes 8 and 10 which are connected to each other (as indicated by the 1s at positions (8, 10) and (10, 8)).\n\nThe query is asking whether there is a path between node 9 and node 4. Looking at the adjacency matrix:\n- Node 9 has no edges connecting it to any other nodes (the entire row and column corresponding to node 9 are zeros).\n- Similarly, node 4 has no edges connecting it to any other nodes (the entire row and column corresponding to node 4 are zeros).\n\nSince neither node 9 nor node 4 has any connections to other nodes, and there are no intermediate nodes that connect these two nodes, there is no path between node 9 and node 4.\n\nFINAL ANSWER: no\n
>
> **5. Although most benchmarks in the evaluation show that the additional context from visual sketchpad brings benefits, I wonder if there are some cases that the additional context introduces noises and causes errors compared with the baseline model.**
>
> **Answer:** This is a good point. Yes. There are instances where the additional context from the visual sketchpad can introduce errors. For example, the vision expert in our framework such as GroundingDINO, may wrongly annotate or miss bounding boxes. We found that GPT-4o is really good at figuring out if these vision experts are making mistakes, and correcting them during reasoning, as we write in L 288-289. But there are times where GPT-4o is still misled by these noises. We appreciate your feedback and will include this point in the limitations section of the final version.
>
> **6. Typos Line 64: jigwaw -> jigsaw**
>
> **Answer:** Thanks for pointing out the typo. We will fix it in the final version.

---

> > ### Author Response · Authors · 2024-08-11
> >
> > Thank you very much for your time and effort in reviewing our paper! If you have any additional questions for discussion, we would be more than happy to address them. We will make every effort to revise our paper based on the reviewer's feedback and suggestions.

---

> > > ### Comment · Reviewer_jSat · 2024-08-13
> > > **Thank you for the rebuttal**
> > >
> > > Thank you for the rebuttal. I think the rebuttal has addressed most of my concerns.

---

### Official Review · Reviewer_2LyD · 2024-07-11

**Soundness:** 4
**Presentation:** 3
**Contribution:** 2
**Rating:** 5
**Confidence:** 3

**Summary:**

This work proposes Visual SKETCHPAD, a framework designed to incorporate visual reasoning into chain-of-thought and tool-use paradigms. Specifically, a multimodal LLM (Large Language Model) addresses a query by (1) generating a plan, (2) executing an action, (3) updating the current context with the result of the action, and (4) iterating until sufficient information is gathered to answer the query.
The plan can include various types of visual reasoning, such as visual program generation via matplotlib and networkx, leveraging specialist vision models like object detectors, segmentation models, and depth estimation models. It also includes the use of specialized tools such as image overlay or a chess drawing Python library. The main contribution of Visual SKETCHPAD lies in the reuse of intermediate visual outputs as additional observations, which enable effective chain-of-thought reasoning. SKETCHPAD demonstrates significant performance improvements compared to text-only chain-of-thought LLMs across multiple tasks.

**Strengths:**

- *Originality:* Incorporation of visual intermediate steps is a natural extension of chain-of-thought reasoning for multimodal LLMs.

- *Quality:* SKETCHPAD is a generalized framework for extending chain-of-thourgh reasoning with visual artefacts produced by vision models, program execution, etc.  A comprehensive evaluation on existing closed-source and open-source LLMs is provided. Authors additionally conduct a human study on the discrepancy between LLM and human made plans.

- *Clarity:* The manuscript is of high quality and easy to follow and the proposed idea is simple to grasp. Authors include prompt examples on supp that clarify the practical form of the proposed framework

- *Significance:* Authors demonstrate a strong performance increase for LLMs prompted via SKETCHPAD compared to text-only chain-of-thought reasoning. The proposed framework does not require any training or tuning and existing LLMs can directly be used with SKETCHPAD style prompts to solve problems that require the described type of visual reasoning.

**Weaknesses:**

- *Clarity:* The reviewer found the title and general positioning of this work slightly misleading. Visual SKETCHPAD naturally would refer to chain of thought framework with that includes generalised sketching capabilities. Troughout this work sketching soley refers to drawing of auxiliary lines for geometric problems and is only one of the multiple tools that the proposed framework makes use of for visual artefact generation.

- *Originality:* The main concern of the reviewer is with the extent of the novelty of this work. The single differentiating factor compared to standard pipelines is the incorporation of visual artefacts on intermediate CoT steps. Incoorporation of visual reasoning information for LLM reasoning cannot be considered a purely novel proposition (for example [a] or [60]). Even though the proposed approach departs from these methods by iteratively updating the context with newly formed visual artefacts, the reviewer is still skeptical w.r.t the extent of the contribution for NeurIPS standards.

[a] See, Think, Confirm: Interactive Prompting Between Vision and Language Models for Knowledge-based Visual Reasoning. Zhenfang Chen, Qinhong Zhou, Yikang Shen, Yining Hong, Hao Zhang, Chuang Gan

**Questions:**

- Could the authors elaborate further on the extent of their contribution and how their work is different from [a] and [60]?

- Authors could include more details about baselines reported on Table 1. Where these follow a similar text-based CoT pipeline?

**Limitations:**

The limitations were properly addressed by the authors.

---

> ### Author Rebuttal · Authors · 2024-08-06
>
> We appreciate your constructive feedback! We are encouraged that you acknowledge the originality, quality, clarity, and significance of our work. We address your concerns as follows, and hope that they can clarify your concerns, and hope that you can improve the rating after seeing the responses!
>
> **1. Clarity: The reviewer found the title and general positioning of this work slightly misleading. Visual SKETCHPAD naturally would refer to chain of thought framework with that includes generalised sketching capabilities. Troughout this work sketching soley refers to drawing of auxiliary lines for geometric problems and is only one of the multiple tools that the proposed framework makes use of for visual artefact generation.**
>
> **Answer:** We respectfully point out that this is a false statement. For vision problems, our work draws numbers and bounding boxes on objects. This is similar to the human sketching process. We circle things on an image and put numbers on it to help reasoning. Even for math problems, we believe that drawing out math functions and graphs is also a sketching process. We as humans often do that on our sketchpad during math examples.
>
> **2. Origniality: the main concern of the reviewer is with the extent of the novelty of this work. The single differentiating factor compared to standard pipelines is the incorporation of visual artefacts on intermediate CoT steps. Incoorporation of visual reasoning information for LLM reasoning cannot be considered a purely novel proposition (for example [a] or [60]). Even though the proposed approach departs from these methods by iteratively updating the context with newly formed visual artefacts, the reviewer is still skeptical w.r.t the extent of the contribution for NeurIPS standards.**
>
> **Answer:** We again respectfully disagree with the reviewer. Updating the visual context during reasoning is the *key novelty* unique to our work, which brings significant performance gain. Prior works, for example [a], [60], VisProg, and ViperGPT,  only do reasoning on text, and the performance is not great. For example, [a] uses GPT-3 and gets 44.6% on OK-VQA. An earlier work, [b], directly prompt GPT-3 with image captions, get 48% by directly prompting GPT-3 with image captions. In ViperGPT, the work that gets the best zero-shot result, but are still far from supervised state of the art. We have been following this line of work, and find the key thing missing is that the LLM cannot make new plans based on new visual contexts. Our method is the first work in this line of research that greatly outperforms all existing state of the art and unleashes the power of the best multimodal LLMs.
>
> Sidenote: Thanks for bringing up [a]. We will cite it in our final version.
>
> [b] Yang, Z., Gan, Z., Wang, J., Hu, X., Lu, Y., Liu, Z., & Wang, L. (2022). An Empirical Study of GPT-3 for Few-Shot Knowledge-Based VQA. AAAI.

---

> > ### Author Response · Authors · 2024-08-11
> >
> > Thank you very much for your time and effort in reviewing our paper! If you have any additional questions for discussion, we would be more than happy to address them. We will make every effort to revise our paper based on the reviewer's feedback and suggestions.

---

> > ### Comment · Reviewer_2LyD · 2024-08-13
> > **Answer to author rebuttal**
> >
> > Thanks for the rebuttal. The reviewer still disagrees with the general positioning of the paper and the amount of the technical novelty. However, as pointed out in the review, this paper showcases some important insights that can help the community in re-iterating over it. The reviewer maintains the initial rating of 5.

---

### Official Review · Reviewer_nzNU · 2024-07-12

**Soundness:** 3
**Presentation:** 3
**Contribution:** 2
**Rating:** 5
**Confidence:** 4

**Summary:**

This paper studies the problem of using language models to generate code to draw for intermediate reasoning. Particularly, the idea of chain-of-thought is applied to facilitate the reasoning process, such that the auxiliary "drawings" enhance the LM's reasoning ability. The proposed method is tested both on math and vision tasks, showing promising results.

**Strengths:**

- Chain-of-thought (CoT) is introduced in the reasoning process. Moreover, visual elements, such as lines, boxes, marks, and segmentation maps, are used for intermediate reasoning steps. In contrast, most previous works only use text.
- The overall pipeline is reasonable, and good results are achieved on different problems, including math and vision tasks.
- The paper is well-organized and written.

**Weaknesses:**

- The overall idea of using LM to generate code to manipulate images is not new. As the authors pointed out, VisProg is one of the most similar works. Personally, this work could be an incremental work based on VisProg by introducing chain-of-thought and applying multimodal reasoning.

- It is very confusing to use the word "sketches" since the drawings are straight lines, boxes, or marks, which are in fact irrelevant to sketches from my view.

**Questions:**

- Is there any evidence that LMs can change the plan given the intermediate visual outcomes during reasoning?
- Is there an in-context learning for LM-based code generator?
- Is that possible to equipt CoT to VisProg (using the same specialist vision models of this work), thus achieving similar performances on math and vision problems?
- To solve the geometry problem, the proposed model is going to generate auxiliary lines. Is there GT that can be used to quantitatively assess the correctness of the lines?
- In Table 3, VisProg performs poorly and the reason claimed is the errors from vision modules. Why Sketchpad did not suffer from this issue?

**Limitations:**

Adequately discussed.

---

> ### Author Rebuttal · Authors · 2024-08-06
>
> Thanks for your valuable feedback! We are honored that you believe the pipeline is reasonable, achieving good results, and the paper well-written. We address your questions below. Hope that we addressed your concerns, and we would be grateful if you could consider improving the rating after seeing our responses!
>
> **1. The idea is not new. VisProg is a similar work. Personally, this work could be an incremental work based on VisProg by introducing chain-of-thought and applying multimodal reasoning.**
>
> **Answer:** Great to hear that you are also following this direction! We have also followed this direction for a long time, and believe Sketchpad fixes the key pain point of VisProg/ViperGPT. The huge performance gain demonstrates the significance of our work. This work actually starts with the authors carefully investigating ViperGPT's trajectories when solving OK-VQA. We find that many LLM-generated programs are wrong, and the vision tools break frequently. We realize that the LLM can be much more powerful if it can investigate the intermediate visual artifacts during the execution of the programs. We further develop this idea and find that it also applies to math and geometry problems. The key innovation is not about vision tool-use, but about multimodal reasoning: LLMs should think step by step across modalities, just like humans do when they are drawing on sketchpads.
>
> **2. It is very confusing to use the word "sketches" since the drawings are straight lines, boxes, or marks, which are in fact irrelevant to sketches from my view.**
>
> **Answer:** We understand your concerns about our terminology. We use the term ‘sketch’ as a metaphor for the process of humans drawing things on a sketchpad while thinking. We are also open to advice on new terminology!
>
> **3. Is there any evidence that LMs can change the plan given the intermediate visual outcomes during reasoning?**
>
> **Answer:** Yes, there is substantial evidence supporting this.  For example, in the dataset V*,for 26% of the examples, GPT-4o find that GroundingDINO fails to detect small objects, and decides to use a sliding window to do a more careful search. Figure 1 (d) is another good example. After using segmentation, GPT-4o decides to use depth estimation to further confirm the answer. Furthermore, the performance gap between VisProg and our method in table 3 quantitatively shows the difference. VisProg can be viewed as a version in which the plan cannot be changed. And its performance is much lower than ours (e.g., 17% v.s. 86% on MMVP).
>
> **4. Is there an in-context learning for LM-based code generator?**
>
> **Answer:** Yes. For computer vision tasks, all tasks share the same prompt, and the prompt contains 6 in-context examples. For math tasks, there are 5 in-context examples.
>
> **5. Is that possible to equipt CoT to VisProg (using the same specialist vision models of this work), thus achieving similar performances on math and vision problems?**
>
> **Answer: ** In our VisProg experiment, to make a fair comparison, we replaced the LLM to GPT-4o/turbo, and then replaced the vision experts with the ones used by our method. It is way worse than our method, as shown in Table 3. So, the short answer is no, VisProg cannot achieve similar performance. The key difference is that in VisProg, the LLM makes the plan after seeing the textual query, without any visual context. We manually inspect VisProg’s traces, and find that many plans are not reasonable and not robust. For example, GroundingDINO often makes mistakes, and the wrong bounding boxes destroy the VisProg programs.
>
> **6. To solve the geometry problem, the proposed model is going to generate auxiliary lines. Is there GT that can be used to quantitatively assess the correctness of the lines?**
>
> **Answer:** Good question! The short answer is no because it is really hard to generate GT for a geometry problem. There are often multiple solutions for the same question, using different auxiliary lines that get the same correct answer. To evaluate the correctness of the lines, we have to show the reasoning trajectory of GPT-4o to humans and let them decide. As shown in L293-297, humans find that for 80% of the cases, the auxiliary line is reasonable.
>
> **7. In Table 3, VisProg performs poorly and the reason claimed is the errors from vision modules. Why Sketchpad did not suffer from this issue?**
>
> **Answer:** Great question! The reason is that Sketchpad includes visual results as part of the reasoning process, allowing GPT-4 to inspect the results and find the errors from vision modules. For example, GroundingDINO may make mistakes for object detection. In VisProg, since the plan is not changed, there is no way to fix it. For Sketchpad, GroundingDINO will also visualize its results, with bounding boxes drawn around the objects. GPT-4 can easily figure out which boxes are wrong by inspecting the visualized results.

---

> > ### Author Response · Authors · 2024-08-11
> >
> > Thank you very much for your time and effort in reviewing our paper! If you have any additional questions for discussion, we would be more than happy to address them. We will make every effort to revise our paper based on the reviewer's feedback and suggestions.

---

### Official Review · Reviewer_BpTp · 2024-07-28

**Soundness:** 3
**Presentation:** 4
**Contribution:** 3
**Rating:** 7
**Confidence:** 4

**Summary:**

The Visual SKETCHPAD framework integrates sketching capabilities into multimodal language models, enabling them to iteratively draw, plan, and reason using visual artifacts - similar to how humans leverage sketching to facilitate problem-solving. Unlike prior approaches that relied on text-to-image models, Visual SKETCHPAD allows language models to directly create visual elements like lines, boxes, and marks, and can even incorporate specialized vision models to enhance the sketching process. Experiments across a range of math and complex visual reasoning tasks demonstrated that Visual SKETCHPAD substantially improves performance.

**Strengths:**

1. Comprehensive Experiment: Evaluated on a wide range of math and complex visual reasoning tasks and achives decent results, which is good.
2. Good Idea: Demonstrates the value of integrating sketching and visual reasoning capabilities into multimodal language models.
Brings language models closer to how humans naturally think and solve problems using a combination of language and visual artifacts.
3. Well delivery. The paper is well-written.

**Weaknesses:**

More discussion about the robustness (repeatibility) is needed, since a common problem for these commercial LVLMs is the instability. Does the method alleviate the invalid responces? How robust is it? Does multiple re-try helps to improve the performance?

**Questions:**

More explaination about the human evaluation is needed. I noticed the author have included the user study. How exactly the user study is conducted? The author is suggested to elablorate this.

**Limitations:**

Yes

---

> ### Author Rebuttal · Authors · 2024-08-06
>
> Thanks for your kind and insightful feedback! We are honored that you believe Sketchpad is a great idea. We address your questions as follows:
>
> **1. More discussion about the robustness (repeatibility) is needed, since a common problem for these commercial LVLMs is the instability. Does the method alleviate the invalid responces? How robust is it? Does multiple re-try helps to improve the performance?**
>
> **Answer:** This is a great question. Regarding invalid responses, we observe that GPT-4-turbo and GPT-4o rarely produce invalid responses for our datasets. There are a few (<5) examples per dataset blocked by safety shields. For robustness, we set the temperature of decoding to 0 to reduce randomness, but still notice some instability in the OpenAI API. To address this, we conducted 3 runs per task. For instance, on BLINK depth, the GPT-4o baseline has a standard deviation of 1.5%, while with visual sketchpad, it's 1.2%. We will update the final version with mean and variance for 3 runs for each task.
>
> **2. More explaination about the human evaluation is needed. I noticed the author have included the user study. How exactly the user study is conducted? The author is suggested to elablorate this.**
>
> **Answer:** For human evaluation, we present the reasoning steps of GPT-4o (interleaved between text and images) to two human subjects. In the geometry task, we ask the annotators to respond with "yes" or "no" to the question: "Would you draw the same auxiliary lines to answer the question?" For computer vision tasks, we ask: "Are the visual reasoning steps reasonable?" We will include an additional human evaluation section in the appendix to provide more comprehensive details.

---

> > ### Author Response · Authors · 2024-08-11
> >
> > Thank you very much for your time and effort in reviewing our paper! If you have any additional questions for discussion, we would be more than happy to address them. We will make every effort to revise our paper based on the reviewer's feedback and suggestions.

---

### Author Rebuttal · Authors · 2024-08-06

We appreciate all reviewers for their timely and positive feedback. We are encouraged that the reviewers believe visual sketchpad is “a good idea” (Reviewer BpTp), “interesting and effective” (Reviewer jSat), with “originality” and “significance” (Reviewer 2LyD). Also, all reviewers believe that the experiments are “comprehensive” and achieved great results; the paper is “well-written” with “great clarity”. (Reviewer BpTp, nzNU, 2LyD, jSat)  We believe visual sketchpad is a novel and effective framework to fully unleash the power of multimodal LLMs, “bringing LMs closer to how humans naturally think” (BpTp) by solving complex problems with a combination of text and visual artifacts.

We will address each reviewer’s questions below. And we are happy to address any further comments from reviewers.

---

### Decision · Program_Chairs · 2024-09-25

**Decision:**

Accept (poster)

**Comment:**

BpTp considers the problem useful/well motivatated and experiments to be comprehensive, covering diverse reasoning tasks.   They ask about determinism of the response and accuracy of responses and receive a satisfactory response in the rebuttal.  The score is maintained at accept (7)

nzNU recognizes the paper contributes to a research trajectory, and asks whether the CoT contribution could be built upon one such technique (VisProg).  A minor point is made re: the use of the terminology ‘sketch’ in this context.  Some questions are asked regarding the technical exposition and about the ground truth for geometry problems.  In my interpretation there are no serious concerns raised by the reviewer only some questions raised out of interest, and the rebuttal has answered them. The reviewer does not reply to the rebuttal so the final score is borderline accept (5).

2LyD is concerned that introducing the modality of ‘sketch’ i.e. diagrams into the chain of thought is not a novel step over existing chain of thought-like approaches.  They also had some questions on how the work is positioned, partially echoing nzNU.  They specifically ask for clarification of novelty over [See et al] and [60], and on the baseline comparisons.  .  There is some reply given around novelty and performance relative [60] and [a] and it should be noted these papers are in arxiv/unpublished state.  The reviewer does not reply to the rebuttal and the score remains at (5).

jSat considers the framework to be interesting and effective, noting the performance gain and SOTA results.  The main weakness identified is lack of comparison with existing CoT frameworks.  The majority of the questions raised are clarification on image formation and edge case behaviour.  The reviewer notes that ‘most’ of their concerns are addressed but has not amended their score of (5).
This is a borderline paper.  No reviewer has argued against acceptance, but only one reviewer is firmly arguing to accept it. The main deficiency in the paper is the lack of comparison to other published visual CoT methods, as the reviewers flagged.  In my view the paper could do better to motivate/position itself relative these works (also see Chen et al. AAAI 2024) as there is an implication in the motivation that this work is first out of the gate on the topic.  The baseline comparisons requested would be interesting but not all of the techniques discussed by the reviewers are published yet.

On the balance the score is low but I don’t see any critical concerns raised by the reviewers that remain post-rebuttal.  I would consider that the paper is borderline acceptable on that bass and so recommend acceptance.